# Enhanced bacteriostasis and osseointegrative properties of SiRNA-modified polyetheretherketone surface for implant applications

Zhen Liu[1]☯, Libin Yang[2]☯, Yazhuo Ni[3]‡, Keying Chen[3], Qiquan Yan[3], Zhiying Zhao[4], Bo Xu[3], Yaoyang Li[3], Rui Li[3]¤‡*, Jianwen Li[5]‡*

1 Department of Stomatology, Airforce Medical Center PLA, Air Force Medical University, Beijing, People's Republic of China, 2 Department of Stomatology, Shizuishan Second People's Hospital, Shizuishan, Ningxia Province, People's Republic of China, 3 Department of prosthodontics, Tianjin Medical University School and Hospital of Stomatology & Tianjin Key Laboratory of Oral Soft and Hard Tissues Restoration and Regeneration, Tianjin, People's Republic of China, 4 Baodi Hospital, Tianjin Medical University, Tianjin, People's Republic of China, 5 Department of Radiology, Shizuishan Second People's Hospital, Shizuishan, Ningxia Province, People's Republic of China

☯ These authors contributed equally to this work.
¤ Current address: Department of Stomatology, Shizuishan Second People's Hospital, Shizuishan, Ningxia Province, People's Republic of China
‡ YN, RL and JL also contributed equally to this work.
* kqxiufulirui@tmu.edu.cn (RL); ninn9928@163.com (JL)

**Data Availability Statement:** All relevant data are within the paper and its Supporting information files.

## Abstract

Polyetheretherketone (PEEK), bearing an elastic modulus that effectively simulates the innate properties of natural bone, has come into the spotlight as a promising bone substitute material. Nonetheless, the biologically inert nature of PEEK, combined with its insubstantial osseointegration and sterilization capabilities, pose constraints on its clinical application in the realm of implants. RNA interference (RNAi), an effective technique used for gene expression regulation, has begun to be applied in implant surface modification. Herein, siCKIP-1 is securely affixed to the surface of PEEK implants, aided by an antibacterial polyphenol tannic acid (pTAN) coatings, enhancing physiologic osseointegration and inhibiting bacterial infection. This method breakthrough not merely facilitates the convenience, but also multifaceted PEEK implants' refinements. The modified PEEK implants have impressive biocompatibility coupled with a noteworthy degree of antibacterial properties. Meanwhile, modified PEEK implants improved osteogenic differentiation of rat bone mesenchymal stem cells (rBMSCs) and demonstrated excellent osteointegrative properties in rat femur implantation models. Therefore, identifying a new implant material with excellent biocompatibility and biomechanical properties is essential.

## Introduction

With the exponential advancement of oral implantology, dental implants are emerging as a progressively favored choice for the replacement of missing teeth. At present, titanium (Ti),

**Funding:** Corresponding author Rui Li serves as funder in the funding (Natural Science Foundation of Tianjin Municipality 18JCYBJC95500). The funder Rui Li had the role in study design, data collection, decision to publish, preparation of the manuscript.

**Competing interests:** The authors have declared that no competing interests exist.

titanium alloys and ceramic materials are widely used as biomaterials for implants due to their excellent osseointegration ability [1, 2]. However, Ti and Ti alloys might cause aesthetic drawbacks owing to their greyish color, and their metal artifacts are considered disadvantageous since they make it difficult to see the bone-implant contact [3, 4]. Although zirconia implants have aesthetic and imaging benefits, the aging problems and brittleness still limit their clinical application [5]. Furthermore, the elastic moduli of the Ti, Ti alloys and zirconia implants are much higher than bone tissue, which might cause stress shielding, trigger loosening of the implants, result in bone resorption [6, 7]. Therefore, it is vital to find a new implant material with good biocompatibility and biomechanical properties.

Polyetheretherketone (PEEK), a semi-crystalline high molecular biopolymer, has been extensively used in orthopedic and traumatic interventions, including interbody fusion cages and artificial joints [8–11]. PEEK possesses strong biocompatibility, remarkable mechanical features, excellent chemical stability, and desirable transmittance properties. More importantly, as a bone substitute material, PEEK possesses an elastic modulus (3-4GPa) comparable to that of bone tissue (1.38–13.8GPa) [12, 13]. Nevertheless, PEEK is a bioinert material with poor osteoconductive and osteoinductive properties on account of its highly hydrophobic and low bioactive surface [14]. In the realm of dental implants, clinical success heavily relies on osseointegration—the process establishing a direct and functional linkage between the implant's load-bearing surface and living bone tissue [15, 16]. To be better used in biomedical implants, the PEEK implants must have a highly bioactive surface that induces abundant bone formation, therefore generating a continuous contact interface with the surrounding bone tissue. Researchers have conducted numerous strategies to boost the surface bioactivity of PEEK, which are mainly divided into blending modification and surface modification [17, 18]. There is a burgeoning interest in surface modification, given its straightforward application and its ability to retain PEEK's inherent properties [19, 20]. Among them, the surface biofunctional of the implant with the extracellular matrix, polypeptides, and growth factors confer an outstanding bioactivity ability to the implant [21, 22]. As a result, we hope to find a suitable biofunctional modification method to improve the PEEK implants' surface bioactivity.

With the in-depth study of the molecular mechanism of bone construction, bioactive RNA molecules are extensively applied in bone tissue engineering as they have been demonstrated to regulate gene expression at the translational, or post-transcriptional levels [23, 24]. In the past decade, RNA-based therapies utilizing various RNA molecule families, such as small interfering RNA (siRNA), microRNA (miRNA), messenger RNA (mRNA), and Long noncoding RNA (lncRNA) have found beneficial applications within the realm of bone tissue engineering [25]. One notable siRNA molecule, SiCKIP-1, has demonstrated its potential to promote bone formation by effectively and specifically downregulating the target protein expression [26, 27]. In particular, one such target is the casein kinase-2 interaction protein-1 (CKIP-1), which is secreted by osteoblasts and hinders bone formation by impeding BMP2 signaling [28–30]. Therefore, we hypothesized that loading siCKIP-1 onto the surface of PEEK implants can siCKIP-1 expression and thus promote osseointegration around implants.

Since RNA is intrinsically unstable, there is a need for a delivery system that can protect siRNA from serum nuclease and effectively deliver siRNA to cells in target tissues [31]. As a "gold standard" carrier for delivering nucleic acids, PEI is the most widely used cationic polymer carrier [32, 33]. Minh K Nguyen et al. successfully employed PEI to transport siNoggin to target cells, leading to improved osteogenic differentiation of rat bone mesenchymal stem cells (rBMSCs) [34]. Negatively charged siCKIP-1 and positively charged PEI can be complexed through ionic interactions to deliver genes to target cells.

How to bind siRNA/PEI complexes to the PEEK implant surface is still a challenge. Biodegradable polyphenol tannic acid (pTAN) coatings have similar properties to poly dopamine

(PDA) coatings, and it can be implemented by self-polymerization of tannic acid (TA) under similar conditions [35, 36]. As TA is widely distributed in plant tissues, pTAN has the advantages of low cost, abundant resources, and colorless. Besides, many studies have shown that it has a broad range of antimicrobial activity [37]. It has been adhered to a diverse selection of materials, and its surface has reactive chemical groups that can be covalently bonded with PEI [38]. The use of the pTAN coating as a secondary reaction platform is expected to achieve drug loading and enhance the surface antibacterial activity of PEEK implants.

To sum up, siRNA is loaded on the surface of PEEK implants via the pTAN intermediate adhesive layer to realize the surface modification of the implants. In our study, we evaluated the osteogenic and antibacterial qualities of modified PEEK material by using cell and bacterial culture through an in vitro assessment. We also implanted modified PEEK material into the thigh bone of rats to assess its in vivo osseointegration. We hypothesize the modification endowed the PEEK implants with better osteogenic and antibacterial properties, thus creating new possibilities for orthopedic implants.

## Materials and methods

### Preparation of PEI/siCKIP-1 complexes

To improve the differentiation of bone cells, rat CKIP-1 (siCKIP-1) was targeted with siRNA. This siRNA consisted of a sense strand (5′–GGACUUGGUAGCAAGGAAAdTdT–3′) and an antisense strand (5′–UUCCUUGCUACCAAGUCCdT*dT–3′), which were obtained from Sangon Biotech in China. FAM-labeled siRNA was also obtained from Sangon Biotech in China. To create the transfection complexes, 10 μL of siCKIP-1 (20 μM) was mixed with 10 μL of PEI (Yuanye, China). FAM labeled PEI/siRNA were prepared in the same way.

### Sample preparation and modification

The PEEK materials were shaped into two different sizes: a circular sample (10mm × 10 mm × 1 mm) and a cylindrical sample (diameter: 2 mm, length: 6mm). The round samples were first polished using abrasive papers with grits of 400, 1000, 1500, and 2000. The cylindrical implants were first polished with dental rubber polishing wheel. Both types of samples were then subjected to ultrasonic cleaning lasting 2 hours in a sequence of acetone, anhydrous ethanol, and purified water, followed by drying at a temperature of 60˚C. Afterwards, the surfaces were immersed in a pTAN solution (2mg/ml, prepared in 100 mM bicine buffer, and 0.6 M NaCl at a pH of 7.8). The substrates were protected from light and gently shaken overnight. After being immersed in water at 37˚C lasting 48 hours, the resulting samples (named PEEK-pTAN) were subjected to an ultrasonic cleaning process lasting 5 minutes followed by a triple rinse, and subsequently soaked for 24 hours in PEI/siRNA complexes or FAM labeled PEI/siRNA solution at 37˚C. Then, the treated samples underwent three rounds of gentle washing to remove any complexes that were physically adsorbed, resulting in the prepared samples (PEEK-pTAN-siRNA or PEEK-pTAN-siRNA with FAM labeled siRNA).

### Surface characterization

The level of surface hydrophilicity was assessed in various PEEK samples using a contact angle meter (ChongDa, JGW-360A, China). At room temperature, a drop of ultra-pure water (2μL) was placed on the surface of each sample, and the contact angle was measured once it reached stability and photographed with a digital camera. All measurements were repeated three times. The surface morphology of various PEEK samples was analyzed using SEM (ZEISS, Gemini 300, Germany). Before scanning by SEM, the samples were sputtered and gilded in an argon

atmosphere by utilizing a sputtering coater. The PEEK-pTAN-siRNA surface with FAM-labeled siRNA was scanned applying a CLSM (Carl Zeiss, LSM900, Germany) to observe the siRNA loading on the PEEK-PTAN surface. AFM (Bruker Corporation, USA) with an OTE-SPA-R 3 probe was used to measure the sample morphology in non-contact mode at a scanning rate of 1 Hz. Chemical composition of samples was characterized employing XPS (Thermo SCIENTIFIC ESCALAB 250Xi) utilizing an Al Ka monochromatized radiation as an X-ray source (1487.20 eV) with a beam spot size of 500 μm.

## In vitro antibacterial function

*Streptococcus sanguis* (*S.sanguis*) ATCC 10556 and *Staphylococcus aureus* (*S. aureus*) ATCC 25923 were grown on freshly prepared brain heart infusion (BHI) agar plates and allowed to grow for 18 hours at a temperature of 37˚C. Subsequently, the bacteria are gathered and resuspended in $5 \times 10^5$ CFU/ml (CFU, colony forming units) to serve as the primary inoculum. The different PEEK samples were positioned with the modified surface facing upwards in a 24-well cell culture plate, and the blank well was used as the control. Then, 1 mL of the BHI bacterial culture that had been diluted was added to each well. After 24 hours, samples were taken out of the plates and subjected to a thorough wash with PBS for three times. Following this, samples were transferred to new plates where they underwent staining utilizing a kit for live/dead staining (Life Technologies Corporation, CA). Subsequently, CLSM was utilized to procure fluorescence images that helped in distinguishing live bacteria (green staining) from their dead counterparts (red staining).

## Cell culture

Rat BMSCs obtained from Cyagen Biosciences in China were grown in fresh Dulbecco's modified Eagle's medium supplied by Gibco in the USA. This medium contained 10% fetal bovine serum (v/v) and 1% penicillin/streptomycin (both provided by Gibco, USA) at 37˚C with 5% $CO_2$. We change the fresh culture medium every two days. In our experiment, we used the third or fourth generation cells. The cells on the culture plate were assigned as a blank control group. Before inoculating rBMSCs, the samples were sterilized by autoclaving.

## Proliferation capacity of cells

To measure the cell proliferation rate, a Cell Counting Ki-8 (CCK8) from Solarbio in China was used. The cells were seeded onto various PEEK samples at a concentration of 10,000 cells per well. After incubating the rBMSCs for 1, 2, 3, 4, 5, 6, and 7 days, each well was replenished with 20 μL of CCK-8 solution and 200 μL of fresh DMEM, followed by a further 2 hours incubation in a 5% $CO_2$ incubator at a temperature of 37˚C. Finally, an indirect assessment of living cell quantity was done by measuring the optical density (OD) of the discharged formazan dye at a specific wavelength (450 nm), utilizing an enzyme-labeled instrument (BioTek Instruments Inc, Germany).

## Cell uptake of siRNA in rBMSCs

PEEK-pTAN-siRNA samples with FAM-labeled siCKIP-1were positioned in 24-well plates. Then, rBMSCs were seeded into the plates with 24 wells at a density of $1 \times 10^4$ cells per well. After 12, 24, and 48 hours of incubation, the samples were rinsed three times with PBS and then treated with 4% paraformaldehyde (from Solarbio, China) for 10 minutes at room temperature to fix them in place. Following this, they underwent 10 minutes of infiltration with

0.5% Triton X-100 (Solarbio, China). Finally, the nuclei in the samples were stained with DAPI dye (Solarbio, China). Samples were observed by CLSM.

## Cell adhesion and morphology

To examine the cellular actin cytoskeleton on the surface of various PEEK samples, rBMSCs were inoculated on the surface of different samples ($1 \times 10^4$ cells/well). Following a 24 hours culture period, the culture medium was taken out, and the samples underwent a triple rinse process with PBS. Subsequently, the cells were subjected to fixation lasting 30 minutes in 4% paraformaldehyde and then permeabilized with 0.5% Triton X-100 for 10 minutes. Rhodamine B Phalloidin from Cytoskeleton, Inc. in the USA was used to stain the actin cytoskeleton. The utilization of DAPI staining facilitated the visualization of cell nuclei. Observations regarding the morphology of the cells were made employing CLSM.

**Alkaline phosphatase activity assays.** In the plates with 24 wells, cells were seeded onto samples at a concentration of $1 \times 10^4$ cells/ml. 24 hours later, the culture medium was changed into osteoinductive medium for osteogenesis induction, which was composed of 50 μM ascorbate, 100 nM dexamethasone, and 10mM β-glycerophosphate, all from Solarbio in China. At day 7 post-osteoinduction, the staining for alkaline phosphatase was conducted as follows. First, the samples were rinsed three times with PBS, followed by fixation in 4% paraformaldehyde lasting 30 minutes, then staining process was then carried out with the BCIP/NBT Alkaline Phosphatase Staining Kit from Beyotime, China. After incubation at room temperature lasting 2 hours in the dark, the staining solution was removed. Following two rinses with ddH2O, the samples were allowed to air-dry and were then photographed with a camera (Canon300, Japan).

## Alizarin red stain assays

For detecting extracellular matrix mineralization in rat bone marrow mesenchymal stem cells, Alizarin red staining was performed. Following the osteoblast induction for 21 days, the staining was carried out in the same way as outlined in the Alkaline phosphatase activity assays. The samples were treated with 4% paraformaldehyde for 30 minutes to get fixed, and then subjected to staining with a 1% solution of alizarin red S (pH 8.4) from Solarbio, China for a duration of 10 minutes. Then they were washed with ddH$_2$O and photographed by camera.

## Quantitative real-time PCR analysis

Once osteogenic induction of 7 or 14 days was completed, the rBMSCs were gathered, and their RNA was extracted using the TRIzol reagent. To transcribe the extracted RNA into cDNA, the QuantiTect SYBR Green PCR kit from Qiagen, Germany was utilized. Then, the LightCycler 480 II system (Roche, LC480 II, Switzerland) was applied for the qRT-PCR analysis. Relative expression of RNAs was determined by employing the $2^{-\Delta\Delta Ct}$ method, with GAPDH serving as an internal reference. Primer sequences utilized for PCR were illustrated in Table 1.

## Western blot analysis

In 6-well plates, rBMSCs were inoculated on different sample surfaces (1×104 cells per well). The cells on the culture plate were designated as blank controls. When the cells achieved the target density, they were subjected to trypsinization using a 0.05% trypsin (trypsin, Gibco) and washed twice with PBS. Subsequently, the cell suspension was supplemented with RIPA lysis buffer (Solarbio, China) to promote the extraction of total protein from different samples.

**Table 1. Primers sequence used for qRT-PCR.**

| genes | forward sequence | reverse sequence |
| --- | --- | --- |
| BMP-2 | 5′–GAAGCCAGGTGTCTCCAAGAG–3′ | 5′–GTGGATGTCCTTTACCGTCGT–3 |
| ALP | 5′–TATGTCTGGAACCGCACTGAAC–3′ | 5′–CACTAGCAAGAAGAAGCCTTTGG–3′ |
| RUNX2 | 5′–TCCAACCCACGAATGCACTA–3′ | 5′–GAAGGGTCCACTCTGGCTTTG–3′ |
| CKIP-1 | 5′–GAGCTTTCGGGTCGATCTGG–3′ | 5′–GGCTCCCTTGTCTGGTCTTT–3′ |
| GAPDH | 5′–CGTCTTCACCACCATGGAGA–3′ | 5′–CGGCCATCACGCCAGTTT–3′ |

Following protein extraction, the proteins were fractionated via sodium dodecyl sulfate-poly-acrylamide gel electrophoresis (SDS-PAGE) and subsequently electrotransferred onto a poly-vinylidene fluoride (PVDF) membrane (Immobilon-P, Millipore, Billerica, USA). After being blocked with 5% bovine serum albumin (from Sigma-Aldrich, USA) for 1 hour at room temperature, the membranes were then incubated with primary antibodies, including anti-ALP, anti-BMP2, anti-RUNX2, and anti-CKIP-1 (all from Bioss, China) at a dilution of 1:1000 and kept at 4˚C overnight. Following incubation with secondary antibodies (at a dilution of 1:5000, sourced from HuaBio, China) for 1 hour at room temperature, membranes were treated with ECL Western Blotting Analysis System from CWBIO, Beijing, China. To carry out semi-quantitative analysis, ImageJ software was used to measure the OD, with GAPDH being employed as an internal control.

## Immunofluorescence staining

Following osteogenic induction for 7 days, rBMSCs were washed with PBS, followed by fixation 4% formaldehyde lasting 30 minutes. Subsequently, permeabilization was carried out with 0.5% (v/v) Triton X-100 lasting 5 minutes. Then, the cells were blocked with 5% bovine serum albumin (BSA) that had been dissolved in PBS for 30 minutes. After incubation with the primary antibody (anti-ALP, anti-BMP2, anti-RUNX2 and anti-CKIP-1), diluted 1:400, at 4˚C overnigh, the fluorescent coupled second antibody was added and incubated lasting 1 hour at room temperature. Finally, the cell nuclei were stained with DAPI, which was followed by imaging with CLSM.

**Mechanism of PEEK-pTAN-PEI/siCKIP promoting osteogenesis.** As described above, protein blotting was carried out. Primary antibodies used were as follows: anti-Smad 1/5 and anti-pSmad 1/5 (1:1000 dilution, Bioss Antibodies, China).

**Osteogenesis experimentin vivo.** Approval for the use of Sprague Dawley rats (SD rats) in this study was approved by the Institutional Animal Care and Use Committee of Yi Shengyuan Gene Technology (Tianjin) Co., Ltd. All the animal experiments were complied with the guidelines of the Tianjin Medical Experimental Animal Care. The study was conducted in accordance with the Declaration of Helsinki and approved by the Institutional Animal Care and Use Committee of Yi Shengyuan Gene Technology (Tianjin) Co., Ltd. (protocol code YSY-DWLL-2021340 and date of 21.04.2021). Male rats aged 6–8 weeks and weighing between 250-280g were utilized in this experiment, with 5 animals being included in each group. The groups included PEEK group, PEEK-pTAN group, and PEEK-pTAN-siRNA group. Isoflurane was administered by inhalation, and a 1% pentobarbital sodium (50mg/kg) intraperitoneal injection was given to anaesthetize the rats. The left leg of the rat is shaved and sterilized with povidone-iodine, a longitudinal incision is made at the patella, and soft tissue is dissected to expose the femoral condyle. Subsequently, a 2 mm diameter hole is created through the femoral condyle along the longitudinal axis of the bone using a low-speed dental drill, while a large amount of cold saline is used to cool the abrasive site in a timely manner. Finally, each of the 3

sets of implants was placed into the femoral foramen and sutured the muscle tissue and skin in layers. After 12 weeks, the rats were euthanized, the femur site of the rat where the implant was placed, fixed with 4% paraformaldehyde, and experimentally analyzed by a high-resolution micro-CT scanner (SkyScan 1276, Germany) and post-decalcification section staining. During the experimental treatment, the rats were effectively anesthetized to reduce the mental tension and pain of the animals. The entire experimental procedure is aseptically operated, reducing postoperative infections and accidental deaths. The anesthetic drug inhalation asphyxia method was used to euthanize the rats and reduce the pain of the animals.

**Microcomputed tomography analysis.** To assess the formation of new bone around the implant in vivo, Micro-computerized Tomography (Micro-CT) from SkyScan 1276 in Germany was employed. The imaging protocol followed the manufacturer's guidelines, with an aluminium filter of 0.25 mm and an isometric resolution of 7 μm. The X-ray energy settings were set at 55 kV and 200 μA. Acquired images were reconstructed and analyzed employing CTAn and CTvol software (Skyscan). A three-dimensional analysis was conducted using the CTAn program to derive microstructural parameters such as the trabecular bone volume/tissue volume ratio (BV/TV), trabecular thickness (Tb.Th), trabecular separation (Tb.Sp), and trabecular number (Tb.N).

## Histological analysis

Upon completion of micro-CT scanning, specimens were subjected to decalcification, dehydration, paraffin embedding, and sectioning using a microtome specifically parallel to the longitudinal axis of the femur. After deparaffinisation, haematoxylin and eosin (H&E) staining, Masson trichrome staining, Sirius Red and immunohistochemical staining were performed. For immunohistochemistry, anti-OPN, anti-OCN (1:200 dilution; Bioss Antibodies, China) were used as primary antibody, with HRP-conjugated goat anti-rabbit IgG (1:200 dilution; ZSGB-BIO, China) acting as the secondary antibody. The stained tissue sections underwent examination using an optical microscope, specifically the Leica S9i microscope from Germany. Images of each section were taken for analysis.

## Statistical analysis

The data were presented as mean ± standard deviation. Group comparisons were performed utilizing one-way analysis of variance. The SPSS 17.0 software was utilized for statistical analysis. Statistical significance was determined when $P$ values $< 0.05$ (**$P < 0.01$ and *$P < 0.05$).

## Results and discussion

### Surface characterization of PEEK-pTAN-siRNA

Water contact angle (S1 Fig) measurements evaluated the hydrophilicity of the sample surfaces. Compared with pristine PEEK (75.5±0.51˚), pTAN-coated PEEK shows higher hydrophilicity (35.1±0.36˚) attributed to the effective modification with hydrophilic phenol moieties of pTAN. The water contact angle of PEEK-pTAN-siRNA was 40.2±0.22˚, which was comparable to PEEK-pTAN. This change in wettability indicates that siRNA immobilization exerts little impact on the contact angle of the pTAN coating. The hydrophilic surface of implant creates a more favorable environment for osteoblast adhesion and bone growth. Surface modification plays a vital role in subsequently cell adhesion [39]. Modification of the PEEK surface with pTAN coating can increase its hydrophilicity without changing the bulk properties.

SEM, CLSM and AFM were utilized to examine the surface morphology changes of various PEEK samples from different dimensions (Fig 1A–1C). SEM images (Fig 1A) show the

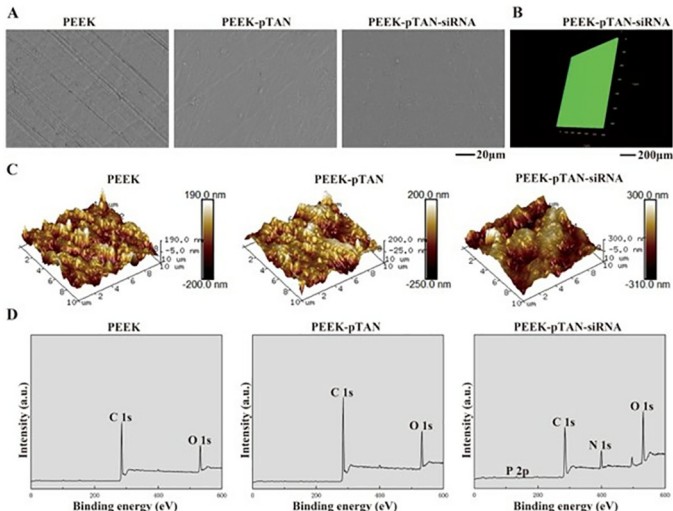

**Fig 1. Surface characterization of PEEK, PEEK-pTAN and PEEK-pTAN-siRNA samples.** (A) Scanning electron microscopy images of PEEK, PEEK-pTAN, and PEEK-pTAN-siRNA. (B) 3D immunofluorescence images of PEEK-pTAN-siRNA with FAM labeled siRNA (siRNA: green fluorescence). (C) AFM images of PEEK, PEEK-pTAN, and PEEK-pTAN-siRNA. (D) XPS spectra of PEEK, PEEK-pTAN, and PEEK-pTAN-siRNA.

relatively flattened and smooth surface of pristine PEEK, while the PEEK-pTAN group exhibits a rough surface morphology with numerous pTAN particles, which confirms the covering of pTAN coating. However, siRNA complex particles, as a small molecule, could not be found on the surface of PEEK-pTAN-siRNA group through SEM. In order to observe the siRNA complex particles attached to the surface of PEEK-pTAN-siRNA, FAM-labeled siRNA was used in this study, and then CLSM was used to observe. Meanwhile, in-depth characterization of surface topography and roughness was carried out by AFM from a 3D perspective. AFM results (Fig 1C) reveal PEEK sample as possessing a remarkably smooth surface (Ra = 39.7 nm). Conversely, the roughness of the PEEK-pTAN (Ra = 48.1 nm) and PEEK-pTAN-siRNA groups (Ra = 76.6 nm) increased, indicating that the surface of the modified PEEK was more was rougher. Thus, it was corresponding to the results of SEM, functionalizing of the PEEK surface significantly affect their surface roughness. Accordingly, this modification method not only enhances the biological efficacy of the PEEK implant surface but also alters its surface topography. Previous studies have shown that the surface topography of implant exerts a fundamental part in the adhesion and proliferation of osteoblasts, with rough surface characteristics offering a superior emulation of the hierarchical structure of bone, thereby promoting advantageous bone formation [40, 41]. Anxiu Xu et al. demonstrated the importance of surface topography for bone formation by creating a rough topography on the peek surface by oxygen plasma and sand blasting [42].

X-ray photoelectron spectroscopy (XPS) is a qualitative tool which is surface-sensitive for characterizing the components of materials. Hence, XPS was performed to further test the surface modification on PEEK. XPS spectrum of PEEK showed obvious C 1s and O 1s peaks, with no evident peaks for other elements (Fig 1D). The carbon to oxygen ratio (4.5) detected in the PEEK-pTAN group exceeded the theoretical C/O ratio (1.65) of pTAN (S1 Table). This discrepancy suggests the presence of significant contamination from carbon-hydrogen compounds, as evidenced by the strong C 1s signal. In the PEEK-pTAN-siRNA group, a distinct nitrogen peak and a smaller phosphorus peak were clearly observed, accompanied by a boost in the N1s and P 2p contents to 12.84% and 1.1% (S1 Table), respectively, confirming the

successful loading of siRNA/PEI complex onto the PEEK-pTAN surface. Taken together, these results suggest that siRNA could decorated onto PEEK-pTAN surfaces to achieve biochemical modification and form a surface with rough morphology and hydrophilicity. It is worth noting that, compared with multilayer self-assembly technology, the surface modification method in this study can realize drug loading only by forming a single layer coating, which is more convenient and time-saving [43]. In contrast to the poly-dopamine coating mediated surface modification, the tannic acid coating is colorless and does not detract from the aesthetics of the implant [3].

## Antibacterial activity of pTAN in vitro

The antimicrobial nature of implants is a necessary design requirement to prevent bacterial infection interfering with osseointegration at the bone-implant interface. Biofilm accumulation can result in an infection that could not be controlled of the peri-implant tissues and ultimately implant failure. Bacterial adherence is the crucial step before bacterial biofilm formation; thus, it is vital to prevent bacterial adhesion onto the implant surface. *S.sanguinis* is a well-established initial colonizer of plaque biofilm and is closely related to peri-implantitis [44]. In addition, *S.aureus*, as a common oral pathogen, is also related to the occurrence of peri-implantitis [45]. In this study, we selected *S.sanguinis* and *S.aureus* to evaluate the antimicrobial effects of pTAN coating modified PEEK. Live and dead bacteria exhibited green and red fluorescence images of various groups in CLSM images, respectively. A low count of dead bacteria (red fluorescence) was observed in both the PEEK group and blank well (Fig 2). While dead bacteria (red fluorescence) of the PEEK-pTAN and PEEK-pTAN-siRNA group increased, indicating that pTAN coating enhanced the antibacterial activity of PEEK. At the same time, the amount of red fluorescence on the PEEK-pTAN and PEEK-pTAN-siRNA group surface decreased, which was caused by the weakening of adhesion ability after the death of bacteria on the sample surface. The antibacterial effect of pTAN was attributed to many antibacterial mechanisms. It has been found that pTAN can inhibit bacterial adherence by inhibiting the enzyme glycosyltransferase. In addition, pTAN can increase the permeability of the bacteria membrane, eventually resulting in the leakage of the intracellular ingredient and bacterial death [37, 46]. Therefore, by forming a pTAN coating with antibacterial effect on

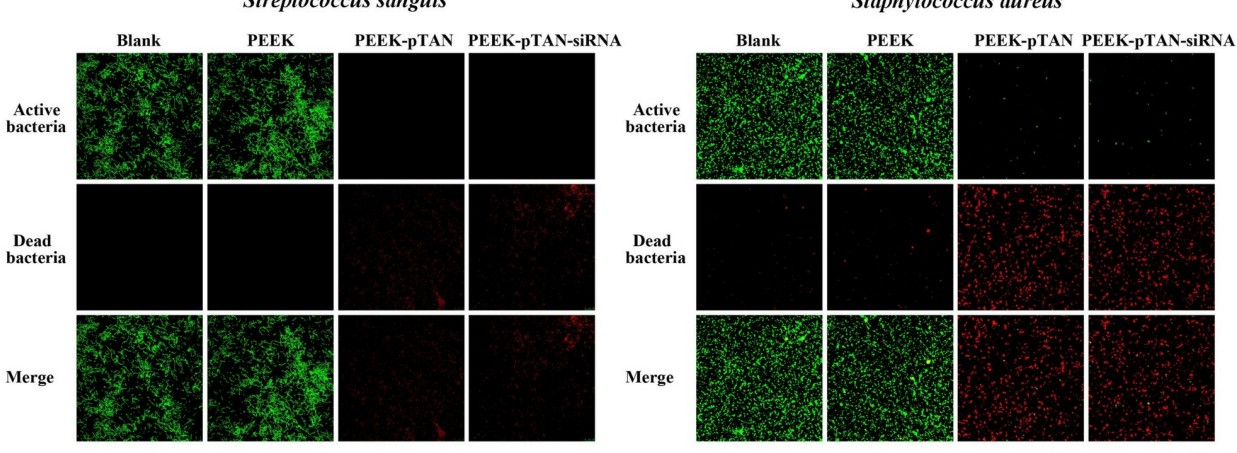

**Fig 2. Live and dead bacteria in CLSM images.** CLSM images of staining after bacteria were cultured on Blank well, PEEK, PEEK-pTAN and PEEK-pTAN-siRNA for 24h: the live bacteria were stained with AO (green), the dead bacteria were stained with EB (red).

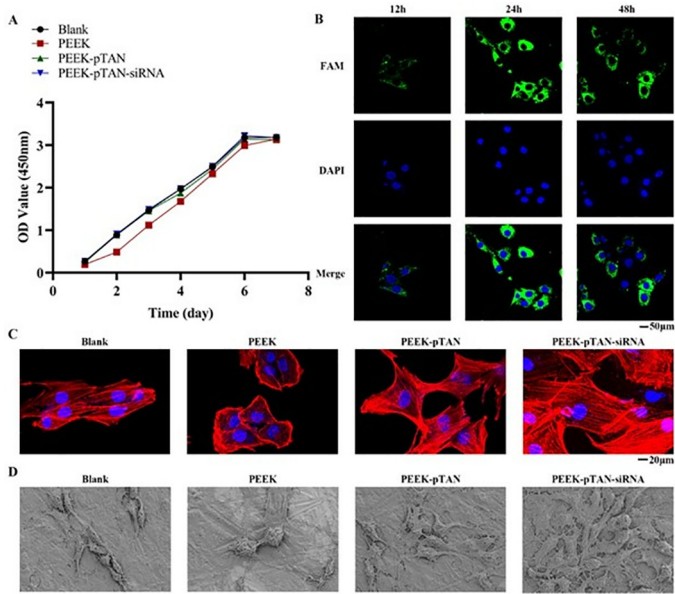

**Fig 3. Biocompatibility evaluation of PEEK, PEEK-pTAN and PEEK-pTAN-siRNA samples.** (A) Proliferation of rBMSC cells on the blank well, PEEK, PEEK-pTAN and PEEK-pTAN-siRNA samples measured by the CCK-8 assay at 1, 2, 3, 4, 5, 6 and 7 days. (B) CLSM of SiCKIP-1 uptake by rBMSC cells at 12, 24, and 48 h. SiCKIP-1 was visualized using FAM (green) and nuclei using DAPI (blue). (C) Cell morphology observed by CLSM after culturing for 24h. (D) Cell morphology observed by SEM after culturing for 24 h.

the surface of peek, the growth and reproduction of bacteria on the surface of PEEK can be effectively inhibited, thus preventing the occurrence of peri-implantitis.

## In vitro biocompatibility of PEEK samples

Cell proliferation and adhesion are considered a paramount step for the ensuing formation of osseointegration [47]. Therefore, to delve into how the modified materials affected the activity of cells in vitro, we conducted the CCK-8, SEM, and CLSM experiments for detection.

CCK-8 cell counting analysis was performed to evaluate the biocompatibility of the modified PEEK. Cells of all samples continued to proliferate over time, eventually reaching a plateau after six days (Fig 3A). Notably, on the second day, the proliferation of rBMSCs in PEEK-pTAN, PEEK-pTAN-siRNA and blank groups was remarkably superior to the PEEK group, while no marked differences were noted among the blank, PEEK-pTAN, and PEEK-pTAN-siRNA groups. This indicates that the biofunctionalized PEEK were noncytotoxic, consistent with previous studies.

SiRNA must be taken up by surrounding cells to exert a regulatory effect in cells [25]. Therefore, we cultured the cells on PEEK-pTAN-siRNA (FAM-labeled siRNA) surface for 48h to evaluate the cellular internalization of siRNA. CLSM ((Fig 3B) identified fluorescence of siRNA was mainly distributed in the cytoplasm and located adjacent to the nucleus. Moreover, the intensity of fluorescence grew progressively with time and peaked at 24 hours, which suggests efficient siRNA translocation into the recipient cells. At 48h, the fluorescence intensity of siRNA declined slightly, demonstrating that siRNA was metabolized in cells.

The adhesion and morphology of rBMSCs seeded on the PEEK's surface and modified PEEK was observed by CLSM and SEM ((Fig 3C and 3D). SEM images showed that rBMSCs was poorly spreading on PEEK and showed a spherical shape with limited filopodia. On the

other hand, cells adhered to PEEK-pTAN and PEEK-pTAN-siRNA showed a long spindle-like shape and stretched out lamellipodia and filopodia which caused cell-cell connection. After culture for 24 h, the cytoskeleton system was visualized using CLSM. In general, cells on PEEK did not show noticeable extension had unremarkable and disorganized actin filaments. Cells in the pTAN and PEEK-pTAN-siRNA groups displayed a well-spread cell morphology with clear, thick actin filaments organized in parallel with the cell's long axis. Especially on PEEK-pTAN-siRNA, cells exhibited better spreading morphology, which substantiated the augmented cell attachment and development of the cytoskeleton.

These results illustrated that modified PEEK has excellent biocompatibility to rBMSCs. Moreover, studies have indicated that surface feature affects cell adhesion and proliferation. Apparently, in this study, PEEK-pTAN and PEEK-pTAN-siRNA groups outperformed those from the PEEK group in terms of cell activity promotion., which highlights the indispensable role of the pTAN coating, which morphologically and hydrophilically alters the material surface, thereby creating the prerequisite conditions for subsequent cell proliferation and differentiation [39, 40, 47].

## Osteogenic differentiation on PEEK-pTAN-siRNA in vitro

Osseointegration is the key to successful dental implant therapy [15]. Osteogenic differentiation of rBMSCs that adhesion on implant surface is critical for obtaining osseointegration [15]. Early osteogenic differentiation is marked by ALP, while alizarin red staining is a common method to identify late osteogenic differentiation of cells [48]. Thus, the osteogenic differentiation potential of rBMSCs cultured with different samples was evaluated through the utilization of Alizarin Red and ALP staining. The PEEK-pTAN-siRNA group exhibited higher ALP activity compared to the PEEK-pTAN group, substantially exceeding that in the PEEK and blank control group (Fig 4A). Consistent with the observation of ALP staining, ARS staining showed increased calcified nodule formation in the PEEK-pTAN-siRNA group compared to the other three groups. The above results suggest that PEEK-pTAN-siRNA surfaces could enhance the differentiation and maturation of rBMSCs into osteoblasts and increase extracellular matrix mineralization, thereby speeding up the formation of new bone.

BMP-2, RUNX2 and ALP are classical osteogenic factors [49, 50]. The qRT-PCR results ((Fig 4B) indicated that the mRNA levels of the above osteogenic factors of PEEK-pTAN-siRNA groups were higher than the PEEK-pTAN group ($P < 0.01$) and remarkably higher than PEEK and blank control groups ($P < 0.01$). Conversely, CKIP-1 is a suppressor of bone formation. The expression levels of CKIP-1 mRNAs of the PEEK-pTAN-siRNA group were the lowest compared to the other three groups ($P < 0.01$). Similarly, the consistent trend was also observed via Western blot analysis ((Fig 4C), indicating that PEEK-pTAN-siRNA sample could decrease the CKIP-1 levels and thus greatly increase the level of osteogenic related factors. Immunofluorescence was then performed to detect the localization of BMP-2 and ALP in cells. It could be seen that the fluorescence intensity of BMP-2 and ALP in PEEK-pTAN and PEEK-pTAN-siRNA groups were noticeably higher than that of the PEEK group, and PEEK-pTAN-siRNA group expressed the most fluorescent signals ((Fig 5A). Correspondingly, these results were further confirmed by immunofluorescence results.

Osteogenic induction ability is essential to peri-implant bone formation and implant osseointegration [51]. The above results show that the PEEK-pTAN-siRNA group had the best osteoinductive ability. BMP-SMAD signaling pathway is a classical way to induce osteoblast development and promote bone formation [52]. Previous studies reported that siCKIP-1 could profoundly stimulate BMP-SMAD signaling pathway [28, 52]. Specifically, CKIP functions as a suppressor of bone formation in BMP-SMAD signaling pathway, which enhances

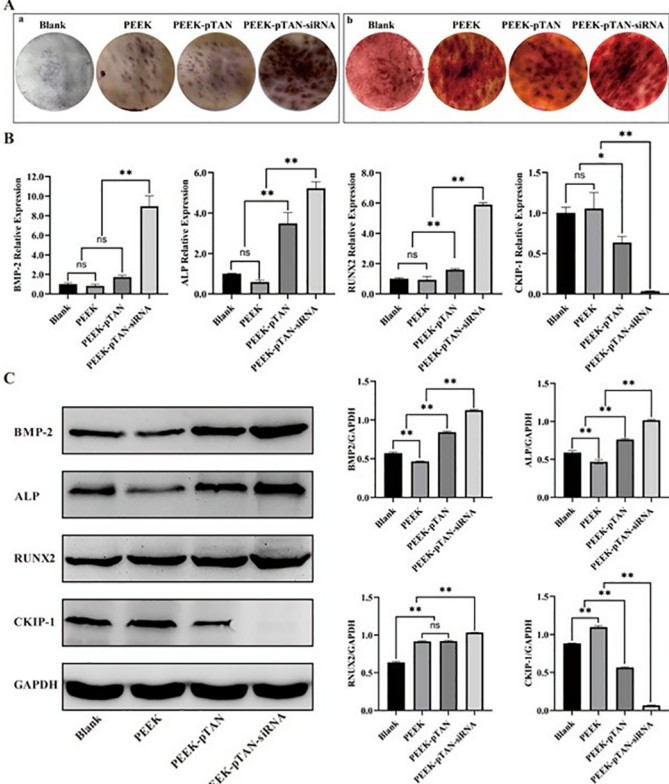

**Fig 4. Evaluation of the ability of PEEK, PEEK-pTAN and PEEK-pTAN-siRNA samples to promote osteogenic differentiation of rBMSCs in vitro.** (A) Staining and quantification of ALP production (a), and ECM mineralization (b). (B) qRT-PCR detection of osteogenic-related genes (BMP2, ALP, RUNX2, and CKIP-1) n = 3, *P < 0.05 and **P < 0.01. (C) Western-blot analysis of osteogenesis-related protein (BMP2, ALP, RUNX2, and CKIP-1) expression.

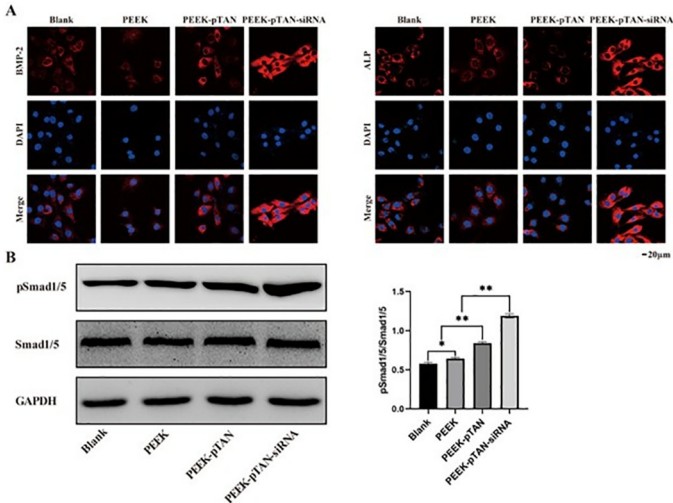

**Fig 5. Evaluation of the ability of PEEK, PEEK-pTAN and PEEK-pTAN-siRNA samples to promote osteogenic differentiation of rBMSCs in vitro.** (A) Immunofluorescence showed the expression of osteoblast-related proteins: BMP2, ALP, RUNX-2, and CKIP-1: red, nucleus: blue. (B) Western blot detection of phosphorylation of Smad1/5. n = 3, *P < 0.05 and **P < 0.01.

the ubiquitin lipase activity of Smurf 1, facilitating the degradation of Smad 1/5, and suppressing the expression of RUNX2 [28, 53]. Moreover, RUNX2 is a major transcription factor in the differentiation of osteoblasts, which regulates osteoblast-specific gene expression (ALP) [54]. SiCKIP-1 selectively targets and eventually degrades complementary mRNA to silence CKIP-1 gene pathway and promote the formation of new bone [25]. Li et al. have shown that the titanium implants loaded with siCKIP-1 on the surface can significantly improve osteoblast differentiation in vitro as well as bone integration in vivo by silencing CKIP-1 expression in the BMP-SMAD signal pathway [55]. In this study, to further confirm the osteogenic mechanism of siCKIP-1, we assess the phosphorylation of signal molecules by Western Blot. In contrast to the other three group, the PEEK-pTAN-siRNA group notably boost the phosphorylation of the Smad1/5/8, indicating that PEEK-pTAN-siRNA regulate bone formation via the BMP2/Smad1/5/8 signal pathway (Fig 5B). Consistent with the previous research results, siCKIP-1 modified PEEK also promoted the osteogenic differentiation of rBMSCs by modulating BMP-Smad signaling pathway.

## Osteogenic differentiation on PEEK-pTAN-siRNA in vivo

An in vivo osseointegration assessment was carried out utilizing a rat femur model in current study. Following a 12h post-operative period, the femurs with implants were extracted and observed by micro-CT scanning with reconstruction to obtain quantitative data related to new bone regeneration and reconstructed 3D images. As the 3D reconstruction and micro-CT images of the samples showed (Fig 6A), abundant newly regenerated bone (yellow) was evident around the PEEK-pTAN-siRNA implants (red), while only minimal new bone was seen around PEEK-pTAN implants, and nearly no new bone formation was observed in the vicinity of the PEEK implants. Furthermore, the quantitative results of the bone volume/total volume (BV/TV), trabecular thickness (TbTh), trabecular number (Tb.N) and trabecular separation (Tb.sp) were shown in Fig 6B. The values of BV/TV, Tb. Th, Tb. N, and Tb. Sp were found to

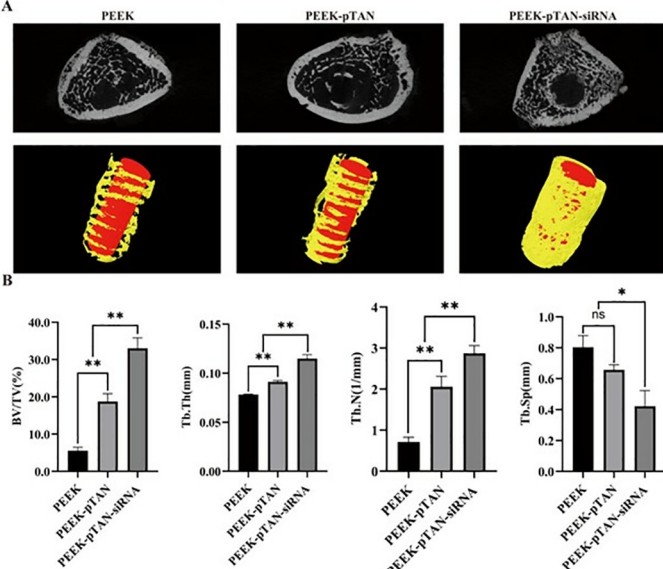

**Fig 6. Evaluation of the ability of PEEK, PEEK-pTAN and PEEK-pTAN-siRNA implants to promote bone regeneration in vivo.** (A) Three-dimensional reconstructed images of samples after implantation for 3 months. (B) Quantitative analysis after implantation for 3 months (trabecular bone volume (BV/TV), trabecular thickness (Tb.Th), trabecular number(Tb.N), trabecular separation (Tb.sp)). n = 3, *P < 0.05 and **P < 0.01.

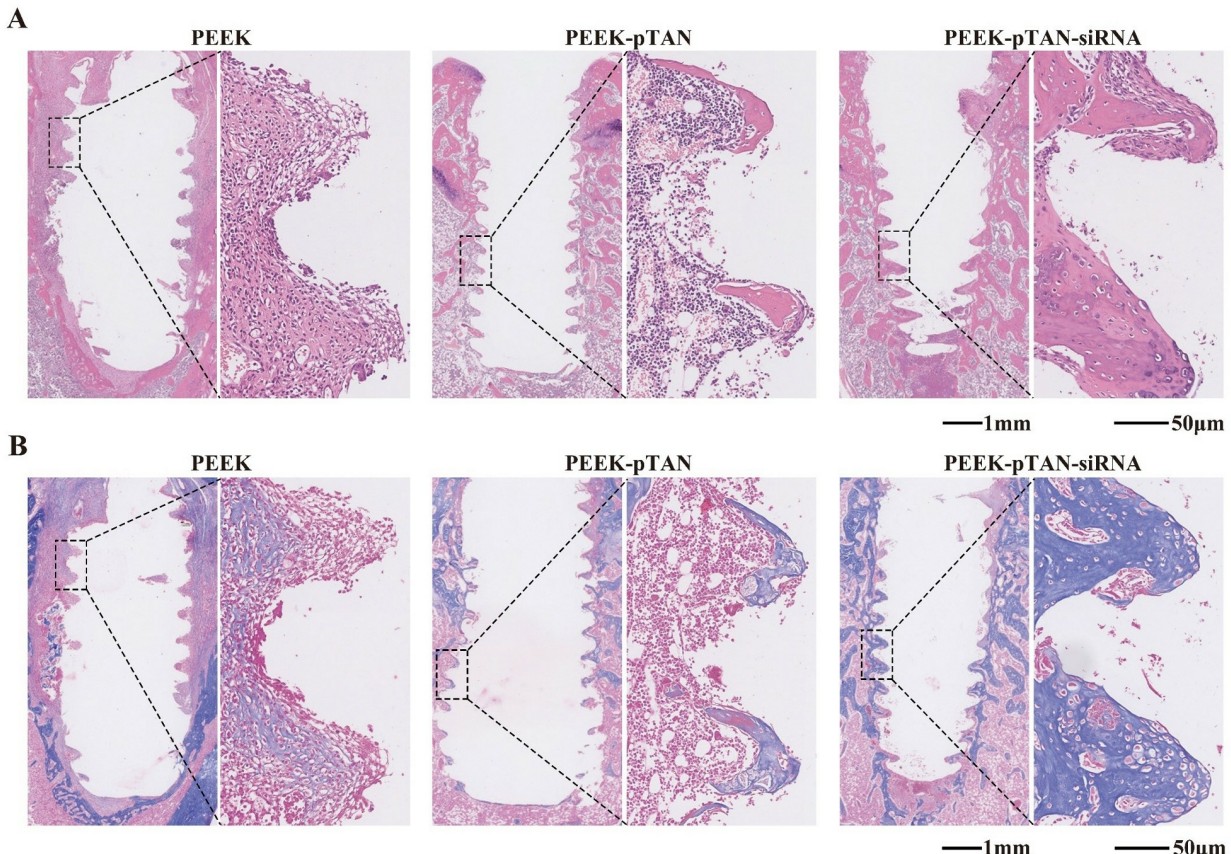

**Fig 7. Evaluation of the ability of PEEK, PEEK-pTAN and PEEK-pTAN-siRNA implants to promote bone regeneration in vivo.** (A) Representative histological sections stained with HE staining after 12 weeks of sample implantation. (B) Representative histological sections stained with Masson staining after 12 weeks of sample implantation.

be the highest in the PEEK-pTAN-siRNA group, followed by the PEEK-pTAN groups, and lowest in the PEEK group ($P < 0.01$). Moreover, the implants with pTAN-siRNA coating have the lowest Tb.Sp ($P < 0.05$).

Histological analysis of decalcified samples was performed by H&E and Masson staining methods to access the progression of collagen and newly formed bone tissue. Collagen secreted by osteoblasts constitutes the bone matrix [56]. In which type I collagen is cross-linked with each other to form a bone matrix framework and eventually mineralized to form normal bone mass [54, 57]. The quality and quantity of collagen keep a certain deposition ratio with mineralization, and it is strongly associated with bone formation [58]. As demonstrated in the Fig 7A and 7B, a large amount of the collagen (the blue-stained area in the Fig 7B) was observed around the implant in PEEK-pTAN and PEEK-pTAN-siRNA group, compared with PEEK groups. The PEEK-pTAN-siRNA group has the richest collagen, which indicates that the new bone formed around implant is the most. The effective performance of the implant in clinical scenarios is largely dependent on the tight contact between the implant surface and the bone tissue, wherein the bone and the implant are in direct contact without the presence of fibrous connective tissue in between [15]. Sirius red staining results show that there was no fibrous tissue component between the implant and the newly formed bone tissue in the PEEK-pTAN-siRNA group, and the bone tissue was in close contact with the implant ((Fig 8A). At the same

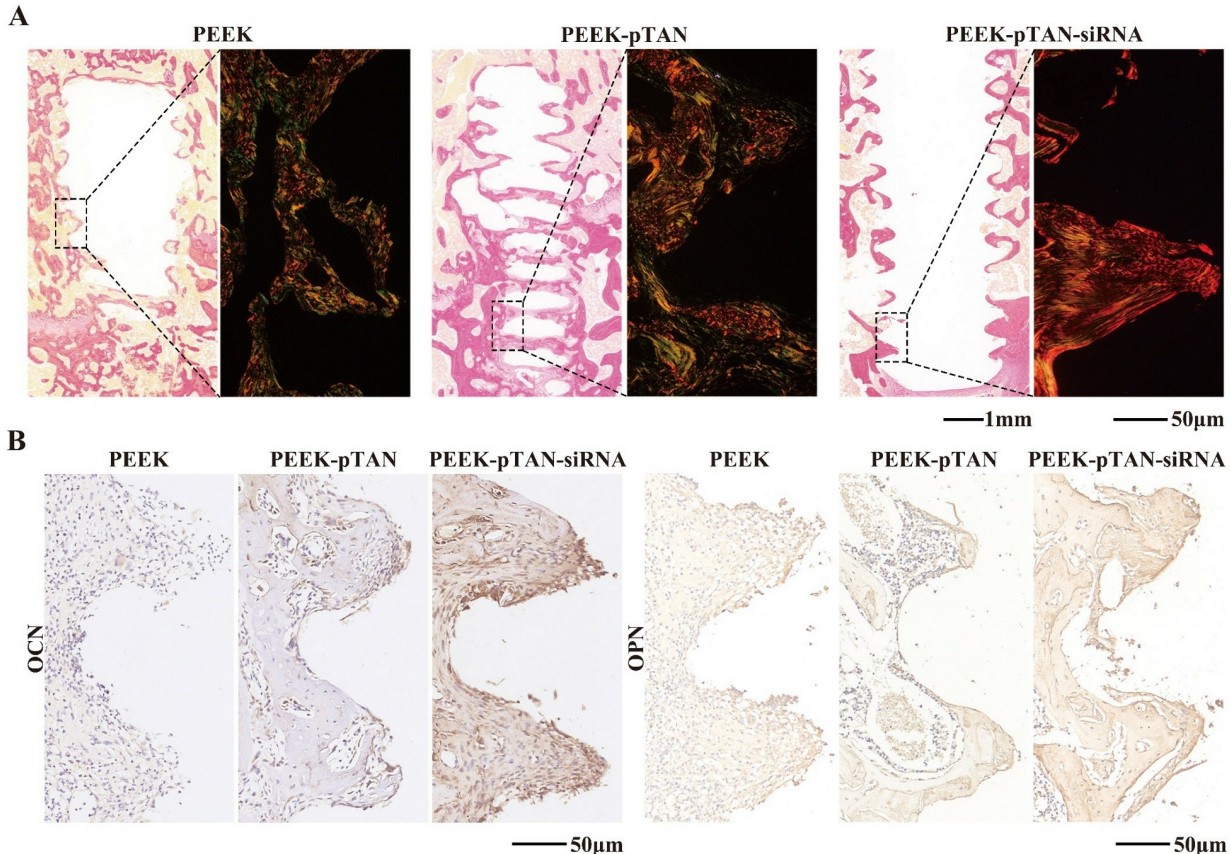

**Fig 8. Evaluation of the ability of PEEK, PEEK-pTAN and PEEK-pTAN-siRNA implants to promote bone regeneration in vivo.** (A) Sirius red staining of PEEK implants surrounding tissues. (B) Immunohistochemical analysis of OCN and OPN expression.

time, no inflammatory infiltration or foreign body reaction was detected by H&E staining results in all groups, demonstrating that the modified PEEK presents excellent biocompatibility in vivo. Immunohistochemical analysis was adopted to examine osteogenic specific proteins expression in the peri-implant tissues ((Fig 8B). Osteopontin(OPN) and osteocalcin (OCN) are extracellular matrix proteins secreted by mature osteoblasts, which exist in the bone marrow and can stimulates the calcification and maturation of bone [59]. In the PEEK-pTAN and PEEK-pTAN-siRNA group, stronger OPN and OCN positive staining was observed than that in the PEEK group. The level of staining was found to be the highest in the PEEK-pTAN-siRNA group. Our results display that the PEEK-pTAN-siRNA implants can effectively facilitate the formation of new bone near the implant. In addition, the in vivo experimental results verified the in vitro experimental findings, which revealed that the pTAN coating improved the implant's biocompatibility by forming a rough hydrophilic implant surface, while siRNA modification endowed the implant with osteoinduction ability. In previous studies, siRNA was loaded on the surface of implants through different surface modification methods, such as cathodic electrodeposition and thermal alkali treatment [60]. Our results demonstrated that siRNA exerted a significant promoting effect on in vivo new bone formation. To sum up, we supposed that the biofunctional modification of siRNA could serve as an efficient approach for modifying the surface of PEEK implants, giving it good cell compatibility in vitro, osteogenic differentiation ability and osteogenic ability in vivo.

## Conclusions

In conclusion, multifunctional modification of PEEK implants surface coating by pTAN coating and siRNA was successfully developed. Osteogenic siRNA was successfully immobilized on the PEEK surface through the antibacterial pTAN layer without changing the bioactive of siRNA. They promote the expression of osteogenic related factors of rBMSCs, inhibit the growth of *S.sanguinis* and *S.aureus*, which endowed PEEK implants the biofunctions of osteogenic and bacteriostasis. This study not only presented a straightforward and effective approach to modify the surface of PEEK implants but also offered experimental evidence supporting their clinical application.

## Supporting information

**S1 Table. Percentages of elements on the surface as estimated by XPS.** Percentage calculations were based on the quantities of C, N, O and P only.
(TIF)

**S1 Fig. Surface characterization of PEEK, PEEK-pTAN and PEEK-pTAN-siRNA samples.**
(TIF)

**S1 Graphical abstract.**
(JPG)

**S1 Raw images.**
(PDF)

## Author Contributions

**Conceptualization:** Zhen Liu, Libin Yang, Yazhuo Ni, Qiquan Yan, Yaoyang Li, Rui Li, Jianwen Li.

**Data curation:** Zhen Liu, Libin Yang, Yazhuo Ni, Zhiying Zhao, Bo Xu, Rui Li, Jianwen Li.

**Formal analysis:** Zhen Liu, Libin Yang, Yazhuo Ni, Keying Chen, Qiquan Yan, Zhiying Zhao, Bo Xu, Yaoyang Li, Rui Li, Jianwen Li.

**Funding acquisition:** Rui Li.

**Investigation:** Zhen Liu, Libin Yang, Yazhuo Ni, Keying Chen, Qiquan Yan, Zhiying Zhao, Bo Xu, Yaoyang Li, Rui Li, Jianwen Li.

**Methodology:** Zhen Liu, Libin Yang, Yazhuo Ni, Qiquan Yan, Rui Li, Jianwen Li.

**Project administration:** Zhen Liu, Libin Yang, Rui Li, Jianwen Li.

**Resources:** Zhen Liu, Libin Yang, Yazhuo Ni, Keying Chen, Zhiying Zhao, Bo Xu, Yaoyang Li, Rui Li, Jianwen Li.

**Software:** Zhen Liu, Yazhuo Ni, Qiquan Yan, Yaoyang Li, Rui Li, Jianwen Li.

**Supervision:** Zhen Liu, Libin Yang, Yazhuo Ni, Zhiying Zhao, Bo Xu, Yaoyang Li, Rui Li, Jianwen Li.

**Validation:** Zhen Liu, Libin Yang, Yazhuo Ni, Qiquan Yan, Yaoyang Li, Rui Li, Jianwen Li.

**Visualization:** Zhen Liu, Libin Yang, Keying Chen, Zhiying Zhao, Bo Xu, Yaoyang Li, Rui Li, Jianwen Li.

**Writing – original draft:** Zhen Liu, Libin Yang, Yazhuo Ni, Rui Li, Jianwen Li.

**Writing – review & editing:** Zhen Liu, Libin Yang, Keying Chen, Zhiying Zhao, Bo Xu, Rui Li, Jianwen Li.

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
