## [Decision Letter · Decision Letter 0]

7 Aug 2024

PONE-D-24-26069Enhanced Bacteriostasis and Osseointegrative Properties of SiRNA-Modified Polyetheretherketone Surface for Implant ApplicationsPLOS ONE

Dear Dr. Li,

Thank you for submitting your manuscript to PLOS ONE. After careful consideration, we feel that it has merit but does not fully meet PLOS ONE’s publication criteria as it currently stands. Therefore, we invite you to submit a revised version of the manuscript that addresses the points raised during the review process.

We look forward to receiving your revised manuscript.

Kind regards,

Rupak Dua

Academic Editor

PLOS ONE

Journal Requirements:

2. To comply with PLOS ONE submissions requirements, in your Methods section, please provide additional information regarding the experiments involving animals and ensure you have included details on  methods of sacrifice and efforts to alleviate suffering.

This study was supported by the Tianjin Key Medical Discipline (TJYXZDXK-038A) Construction Project.

4. In the online submission form, you indicated that Supporting material is available from the corresponding author (kqxiufulirui@tmu.edu.cn) on reasonable request.

. 

7. Please remove your figures from within your manuscript file, leaving only the individual TIFF/EPS image files, uploaded separately. These will be automatically included in the reviewers’ PDF.

8. PLOS ONE now requires that authors provide the original uncropped and unadjusted images underlying all blot or gel results reported in a submission’s figures or Supporting Information files. This policy and the journal’s other requirements for blot/gel reporting and figure preparation are described in detail at https://journals.plos.org/plosone/s/figures#loc-blot-and-gel-reporting-requirements and https://journals.plos.org/plosone/s/figures#loc-preparing-figures-from-image-files. When you submit your revised manuscript, please ensure that your figures adhere fully to these guidelines and provide the original underlying images for all blot or gel data reported in your submission. See the following link for instructions on providing the original image data: https://journals.plos.org/plosone/s/figures#loc-original-images-for-blots-and-gels.   

Reviewers' comments:

Reviewer's Responses to Questions

**Comments to the Author**

1. Is the manuscript technically sound, and do the data support the conclusions?

Reviewer #1: Yes

Reviewer #2: Yes

Reviewer #3: Yes

2. Has the statistical analysis been performed appropriately and rigorously? 

Reviewer #1: Yes

Reviewer #2: Yes

Reviewer #3: Yes

3. Have the authors made all data underlying the findings in their manuscript fully available?

Reviewer #1: Yes

Reviewer #2: Yes

Reviewer #3: Yes

4. Is the manuscript presented in an intelligible fashion and written in standard English?

Reviewer #1: Yes

Reviewer #2: Yes

Reviewer #3: Yes

5. Review Comments to the Author

**Reviewer #1: **I have attached the comments in a word file and have uploaded it here.

My recommendation is based on the comments in the attached file and, I recommend that the manuscript be revised and resubmitted.

**Reviewer #2:** The title could be made more concise. For example, "Enhanced Bacteriostatic and Osseointegrative Properties of SiRNA-Modified Polyetheretherketone for Implant Applications." The abstract should include the main statistical tests used. In the introduction, a suggested improvement: "Therefore, identifying a new implant material with excellent biocompatibility and biomechanical properties is essential." Clarify the statistical methods used for data analysis in the Materials and Methods section. Ensure all figures in the Results section have complete and informative captions. Expand on the limitations of the study in the Discussion section, and ensure the conclusion highlights the study's significance and potential impact.

**Reviewer #3: **This is a nicely written paper with a comprehensive set of data including in vitro and in vivo studies. The authors provide detailed surface characterization results. Then, they provide comprehensive in vitro results with bacteria as well as mammalian cells. Moreover, the authors also provide comprehensive in vivo data showing promising results. A lot more can be done but the results presented are in a good manner.

6. PLOS authors have the option to publish the peer review history of their article (what does this mean?). If published, this will include your full peer review and any attached files.

Reviewer #1: No

Reviewer #2: No

Reviewer #3: No

---

## [Author Response · Author response to Decision Letter 0]

25 Sep 2024

Dear reviewers:

On behalf of my co-authors, we thank you very much for giving us an opportunity to revise our manuscript, we appreciate the editor and reviewers very much for their positive and constructive comments. We have revised the whole manuscript according to the journal and reviewers' requirements and would like to re-submit it for your consideration. All revisions in the manuscript have been identified by using the red font. We have already addressed the comments raised by the reviewers. Below, we have listed our responses to each of the reviewers' comments, point by point.

We hope the revised version of the manuscript is acceptable for publication in your journal now. Looking forward to hearing from you.

Another noteworthy change is that the funding statement is not correct, we haved amend it. Corresponding author Rui Li serves as funder in the funding (Natural Science Foundation of Tianjin Municipality 18JCYBJC95500). The funder Rui Li had the role in study design, data collection, decision to publish, preparation of the manuscript.

Yours sincerely,

Rui Li

Corresponding author: Rui Li

E-mail: kqxiufulirui@tmu.edu.cn

---

## [Decision Letter · Decision Letter 1]

6 Nov 2024

Enhanced Bacteriostasis and Osseointegrative Properties of SiRNA-Modified Polyetheretherketone Surface for Implant Applications

PONE-D-24-26069R1

Dear Dr. Li

We’re pleased to inform you that your manuscript has been judged scientifically suitable for publication and will be formally accepted for publication once it meets all outstanding technical requirements.

Kind regards,

Rupak Dua, Ph.D. 

CEO, Materials Metric LLC

Academic Editor

PLOS ONE

Additional Editor Comments (optional):

Reviewers' comments:

Reviewer's Responses to Questions

**Comments to the Author**

1. If the authors have adequately addressed your comments raised in a previous round of review and you feel that this manuscript is now acceptable for publication, you may indicate that here to bypass the “Comments to the Author” section, enter your conflict of interest statement in the “Confidential to Editor” section, and submit your "Accept" recommendation.

Reviewer #1: All comments have been addressed

2. Is the manuscript technically sound, and do the data support the conclusions?

Reviewer #1: Yes

3. Has the statistical analysis been performed appropriately and rigorously? 

Reviewer #1: Yes

4. Have the authors made all data underlying the findings in their manuscript fully available?

Reviewer #1: Yes

5. Is the manuscript presented in an intelligible fashion and written in standard English?

Reviewer #1: Yes

6. Review Comments to the Author

Reviewer #1: My all comments have been addressed in the resubmitted manuscript. Therefore, kindly accept the manuscript as is.

7. PLOS authors have the option to publish the peer review history of their article (what does this mean?). If published, this will include your full peer review and any attached files.

Reviewer #1: No

---

## [Editor Report · Acceptance letter]

22 Nov 2024

PONE-D-24-26069R1 

PLOS ONE

Dear Dr. Li, 

I'm pleased to inform you that your manuscript has been deemed suitable for publication in PLOS ONE. Congratulations! Your manuscript is now being handed over to our production team.

Kind regards, 

on behalf of

Dr. Rupak Dua 

Academic Editor

PLOS ONE